# Rational Design of a Portable Chemometric-Assisted Voltammetric Sensor Based on Ion-Imprinted Polymeric Film for Co(II) Determination in Water

**DOI:** 10.3390/nano14060536

**Published:** 2024-03-18

**Authors:** Sabrina Di Masi, Nelson Arturo Manrique Rodriguez, Marco Costa, Giuseppe Egidio De Benedetto, Cosimino Malitesta

**Affiliations:** 1Laboratory of Analytical Chemistry, Department of Biological and Environmental Sciences and Technologies, University of Salento, 73100 Lecce, Italy; nelsonarturo.manriquerodriguez@studenti.unisalento.it (N.A.M.R.); marco.costa@unisalento.it (M.C.); 2Laboratory of Analytical and Isotopic Mass Spectrometry, Department of Cultural Heritage, University of Salento, 73100 Lecce, Italy; giuseppe.debenedetto@unisalento.it

**Keywords:** Co(II), sensor, imprinted polymers, electropolymerisation, chemometry, Taguchi experimental design, Langmuir–Freudlich isotherm

## Abstract

Herein, chemometric-assisted synthesis of electrochemical sensors based on electropolymerised ion-imprinted polymeric (e-IIP) films was explored. Co(II)-IIPs sensors were prepared by performing electropolymerisation procedures of polymerisation mixtures comprising varying concentrations of an electroactive o-aminophenol (o-AP) monomer and Co(II) ions, respectively, according to the Taguchi L9 experimental design, exploiting the simultaneous evaluation of other controlled parameters during electrosynthesis. Each e-IIP developed from Taguchi runs was compared with the respective non-imprinted polymer (NIP) films and fitted according to Langmuir–Freudlich isotherms. Distinctive patterns of low and high-affinity films were screened based on the qualities and properties of the developed IIPs in terms of binding kinetics (KD), imprinting factor, and the heterogeneity index of produced cavities. These results can provide a generic protocol for chemometric-assisted synthesis of e-IIPs based on poly-o-AP, providing highly stable, reproducible, and high-affinity imprinted polymeric films for monitoring purposes.

## 1. Introduction

Intense anthropogenic activities continuously contribute to the pollution of agriculture fields and water bodies in general, with a particular focus on emissions of pollutants which influence biological ecosystems’ health by causing adverse effects. Among pollutants, heavy metal ions (HMIs) remains of a major concern, and in particular, trace HM ions play an important role in environmental contamination due to their toxicity and potential mobility.

Cobalt (Co) is a trace element that accumulates in the environment as a result of both natural and anthropogenic activities. In industrial wastes, the concentration of Co ions originates from a range of metallurgic processes (such as those involved in catalysts and batteries). In surface waters, Co concentration naturally occurs at concentration of µg/L [1]. Biologically, Co is an important constituent of vitamin B12, but elevated concentrations can be potentially involved in cardiac, respiratory, and carcinogenic effects on human health [2,3,4]. However, water quality standards for cobalt have not been established for the European Union yet. The current quantification limit of Co(II) in fresh water is set to 0.05 µg/L, whereas 2 µg/L is reported in marine water [5].

Traditionally, the monitoring of Co(II) ions has been performed using conventional techniques, such as chromatographic techniques [6], spectrophotometry [7], fluorescence [8], inductively coupled plasma [9], flame atomic absorption spectrometry [10], and voltammetry [11]. The last electrochemical method is widely employed in the field of electrochemical sensors, since it provides accurate and sensitive in situ determination of plenty of analytes in miniaturised devices [12,13].

Among sensitive and selective receptors adopted in electrochemical sensors, molecularly imprinted polymers (MIPs) have attracted great attention. MIPs can be customised for a specific molecule of interest by tailoring the structure of polymers during synthesis. Hence, the monomers are mixed with the selected target (template) during polymerisation and then removed with a suitable solvent. The result is an imprinted polymer with cavities that mimic the shape of the target. This affinity allows the MIP to selectively attract the target molecule, making it more efficient at adsorbing the molecule with improved selectivity and specificity. 

The synthetic materials used in MIPs cover a broad spectra of applications [14], the most-known ones being used in solid-phase extraction procedures [15] and in sensing applications for selective recognition of analytes and related monitoring [16,17]. Rational design of MIPs requires controlling parameters affecting the synthesis procedure, which also greatly affect the properties of the resulting polymer in terms of affinity capabilities and the imprinting factor (IF) (which is a parameter used to compare MIPs with their counterpart, not-imprinted polymers, NIPs). Optimisation based on chemometric techniques (screening and response surface methods) can be advantageously applied to improve analytical features of the developed MIPs, allowing for the simultaneous evaluation of optimal synthesis parameters in a multilevel space. However, few chemometric applications for enhancing quality properties of MIPs [18,19] have been reported.

Our research activities have already focused on sensor preparation based on MIPs for HM ion monitoring in water [20,21,22,23]. We have also explored electropolymerisation procedures in obtaining ion-imprinted polymeric films (e-IIPs) directly attached to transducer elements of screen-printed electrodes, allowing for practical application of disposable sensing units for in situ applications. In this work, we propose a chemometric-assisted preparation of high-affinity, -imprinted factor, and -selectivity electropolymerised ion-imprinted polymeric films (e-IIPs) for Co(II) ions’ determination in aqueous samples. To the best of our knowledge, this is the first report combining chemometric methods of Taguchi orthogonal experimental design and Langmuir–Freudlich binding isotherms for data sensor analysis, both of which able to utilise and predict quality properties of MIP-based sensors during optimisation problems.

Therefore, the developed Co(II)-IIP sensor was electrochemically characterised to studying its selectivity, reproducibility, and reusability features. Cross-reactivity studies against other metal ions such as Ni(II), Zn(II), Cr(III), Hg(II), Mn(II), and Cd(II) are reported here. Recovery tests in spiked tap and sea water samples are also reported, demonstrating the potential ability of the proposed sensor to recognise Co(II) ions in complex matrices.

## 2. Materials and Methods

### 2.1. Chemicals

MES hydrate, o-Aminophenol (o-AP, 99%), cobalt nitrate hexahydrate (98%), cadmium nitrate tetrahydrate, mercury(II) nitrate monohydrate, nickel(II) nitrate hexahydrate, manganese(II) nitrate hydrate, zinc nitrate, and chromium(III) nitrate nonahydrate were purchased from Merck (Milano, Italy). Glacial acetic acid and sulphuric acid (95%) were commercially available as analytical grade reagents (VWR, Milano, Italy). All reagents were used without further purification. Unless otherwise specified, the chemicals used in the experiment were of analytical grade and the water was double-deionised water (Milli-Q) used for washing the polymeric film after preparation and to prepare the buffered solutionsGlacial acetic acid and NaOH were mixed to obtain Acetate Buffer (AB) 50 mM at pH 5.3 and used as the media during the electropolymerisation. MES buffer (50 mM, pH 5) was used to prepare stock solution of Co(II) and the related diluted solution.

### 2.2. Apparatus

Cyclic voltammetry (CV) and electrochemical impedance spectroscopy (EIS) measurements were made using a PalmSens potentiostat equipped with a cable connector (DropSens, Milano, Italy) for screen-printed electrodes. PSTrace software v. 5.9 was used to control the instrument and for data acquisition. The polymeric film was deposited on screen-printed carbon electrodes (SPCEs). The SPCEs were commercially available (DRP-150, Metrohm, Milano, Italy). The SPCEs were composed of a three-electrode configuration on a planar ceramic support (3.3 cm × 1 cm), and they consisted of a carbon disk-shaped working electrode (4 mm diameter), a platinum counter electrode, and a pseudo-Ag reference electrode, separately.

### 2.3. Preparation of Sensors Based on Co(II)-IIP and NIP Films

Co(II)-IIP films and NIP films were fabricated by means of electrochemical polymerisation. Briefly, a bare SPCE was previously subjected to CV measurements for 5 consecutive scans in MES buffer (pH 5). Then, it was immersed in the electropolymerizing solution, which was a pH 5.3 acetate buffer (AB, 50 mM) containing variable mmol/L of o-AP, and variable mmol/L of Co(II), separately. Cyclic voltammetry of polymeric film growing was applied between −0.2 V and 1.2 V vs. pseudo-Ag. After the modified electrode with Co(II)-IIP films was obtained, it was gently rinsed with MilliQ to remove unreacted monomers. Elution of the target was operated by immersion of the unwashed Co(II)-IIP films in 0.3 mol/L of H_2_SO_4_ for variable minutes (see Section 2.4). Thereafter, the obtained Co(II)-IIP film sensors were rinsed with plenty of MilliQ water and were ready to be used. The NIP film was prepared exactly the same as above, except that no Co(II) was present in the electropolymerizing solution.

### 2.4. Taguchi Experimental Design

Multivariate optimisation was performed to optimise the development of Co(II)-IIP and NIP films. A Taguchi orthogonal L9 (3^4) array experimental design was selected to optimise the sensor preparation at three levels (Table 1) of four main affected variables: (i) electroactive monomer concentration, (ii) target concentration, (iii) CV during electropolymerisation, and (iv) time of elution. All of the selected variables are potentially involved in the formation of cavities on the polymeric films and their effective availability for rebinding purposes. Levels for each variable were chosen on the basis of specific ratios between the monomer and target that can coordinate differently in the different applied conditions. Minitab^®^ software (version 18) was used for the design, mathematics modeling, and optimisation.

The experimental design was carried out according to the criterion “larger-the-best” with the aim to evaluate the maximum imprinting factor (IF) and sensitivities of the developed sensors (Co(II)-IIP compared to NIP films). Affinity properties of the formed cavities (K_D_) and imprinting factors (IF = *i*_max_ IIP/*i*_max_ NIP) for the developed sensors were obtained from Langmuir–Freundlich (LF) isotherms and used as Y_1_ and Y_2_ responses in the Taguchi experimental design, separately. The signal-to-noise ratio (S/N ratio) of the responses was used to indicate the magnitude of response change to variations in the controlled parameters, and finally to achieve optimal conditions of sensor design. 

### 2.5. Co(II) Ion Sensing

Electrochemical responses of Co(II)-IIP and NIP films toward Co(II) ions were recorded by DPV measurements. Basically, after the electropolymerisation process, a reduction peak potential of −0.3 V vs. pseudo Ag was visible. After the wash treatment, this was slightly shifted at −0.38 V, which was attributed to the electrochemical reduction of poly-o-aminophenol at the carbon surface electrodes. The redox behaviour of the obtained polymer was tested after subsequent rebinding of Co(II) ions during the calibration test (drop-cast incubation for 5 min), providing a progressive reduction in the DPV peak current with increased tested concentrations of the target (0 blank, 1.95, 3.8, 7.16, 15.35, 61.5, and 125 nM). Analytical performances were assessed by triplicate DPV measurements at the different concentrations. Sensor responses were obtained as Δ*i* (current peak at blank subtracted from current peak acquired at known Co(II) concentration). Langmuir–Freundlich isotherms (performed using OriginPro 16) were used to obtain non-linear calibration curves for developed Co(II)-IIP and NIP film sensors, separately. From each fitting, the dissociation constant (K_D_), binding capacity (*i*_max_), and heterogeneity factor (n) were extracted and used for comparison results. Linear regression plots at lower concentrations of Co(II) ions were used to calculate LOD and sensitivities of Co(II)-IIP film sensors.

### 2.6. Characterisation of Sensors

The electrochemical characterisation of Co(II)-IIP films and NIP films was carried out by CV and electrochemical impedance spectroscopy (EIS) measurements after each electrode modification. Very briefly, in the case of CV measurements, the characterisation of electrode modification was studied in the presence of MES buffer (5–10 CV scans). EIS measurements were obtained in 0.1 M KCl containing a 10 mM Fe(CN)_6_^3−/4−^ redox couple for bare SPCEs, after the electrosynthesis, after the wash treatments, and after 10 min of Co(II) ion solution incubation for each studied electrode. CVs were set from −0.7 to +0.5 V, with 0.01 potential step and 0.1 mV/s scan rate, whereas EIS measurements were accomplished at +0.2 V as a fixed potential, from 10 kHz to 0.1 Hz and the applied frequencies (49 points per decade), 0.25 V as the potential amplitude, and 10 s as the time of equilibration. Nyquist experimental data were fitted according to a Randles circuit (see Appendix A). The EIS measurements were carried out in triplicate at room temperature.

### 2.7. Real Sample Analysis

In order to test the applicability of the proposed sensor to real aqueous samples, three matrices were chosen, such as commercial drinking water (pH 7.7), tap water from our laboratory (pH 7.3), and sea water (pH 8.2) samples. To do this, prior to analysis, the real samples were buffered at pH 5 (1:10 MES buffer) and used as media baseline (blank). The obtained solutions were allowed to rotate under stirring for 30 min. Thereafter, the buffered samples were spiked with a known amount of Co(II) ions. Finally, 40 µL of the spiked samples was drop-casted onto the Co(II)-IIP film sensor to record the sensor signal in the spiked samples for comparison with a standard solution.

## 3. Results and Discussions

### 3.1. Preliminary Tests

To investigate the feasibility of developing Co(II)-IIP film sensors for the sensitive determination of Co(II) ions, a preliminary test on a prepared non-optimised Co(II)-IIP film sensor was conducted and is here discussed. Briefly, the development of sensors started by adopting the following conditions: (i) monomer–target concentrations were 0.5–2 mM; (ii) CV scans during sensor synthesis were kept constant at 10 cycles to promote the formation of the polymeric films; (iii) 10 min of elution in 0.3 M H_2_SO_4_ was chosen to remove the target from the polymer backbone and to create cavities. Figure 1a shows the preliminary assessment of five consecutive concentrations of Co(II) ions and the related calibration plot (Figure 1b).

As seen from Figure 1a, in the selected buffer of MES at pH 5, a reduction peak potential is visible at −0.4 V vs. Ag. The observed potential peak can be attributed to the behaviour of the formed polymeric structure that can be modulated after the interaction with Co(II) ions. In fact, we observed a decrease in its current density with increased tested concentrations of Co(II) ions. This result agrees with that observed during the preparation of the sensors compared to the NIP films, which reported a modulated mechanism of polymer electroactivity upon the presence of Co(II) ions during polymer growing. The washing treatment in acidic conditions (0.3 M H_2_SO_4_) plays a role in the destruction of hydrogen bonds between the target and polymer based on o-AP [24,25].

### 3.2. Optimisation of Sensor Performances

The imprinting process of IIPs is generally governed by the simultaneous contributions of various applied synthesis conditions, which are also involved in obtaining the best performance recognition capabilities at the cavities. Here, the ability of the developed polymer to coordinate and sensitively recognise the Co(II) ions was explored by means of experimental trials modelled on the Taguchi experimental design. This design was used to find the optimal conditions for Co(II)-IIP film sensor preparation with respect to sensor affinity towards Co(II) ions and imprinting factor (IF), compared to those of NIP films. The orthogonal L9 (3^4) array design arranged each studied variable into the selected levels by combining both Y responses of each sensor (Y_1_ = K_D_ and Y_2_ = IF, separately). Appendix A reported the computed S/N ratios of the Y_1,2_ responses. The contribution of each variable to the Y responses can be determined by comparing the calculated max – min (∆) values in the response table for the S/N ratios (Appendix A). A high ∆ value means a more effective contribution to the response variation, which results in the following order: elution time > [Co(II)] > [o-AP] > CV scan numbers. The computed S/N ratios calculated according to the “Larger the best” criterion showed that the optimal conditions are as follows: 0.5 mM o-AP, 3 mM Co(II) concentration in the polymerisation mixture, 30 CV scans during the electrosynthesis, and 10 min of elution time in 0.3 M H_2_SO_4_ for obtaining the maximisation of both the sensor affinity towards Co(II) and the imprinting factor to effectively recognise the target.

Hence, the kinetics parameters (Appendix A) showed spatial differentiation of the IIPs according to K_D_, n, and Δ*i_MAX_,* suggesting that the presence of Co(II) in the polymerisation mixture modulates the quality properties of the developing cavities in terms of their effective recognition of and binding to the target. From the results shown in Appendix A, it can be noted that NIP sensors show better performance in terms of measurement repeatability compared to different sensors. At the same time, higher sensitivities are visible for the Co(II)-IIP film sensor, exhibiting poor but explained variability between each measurement. This could be related to the different patterns of IIPs, suggesting a differentiation of the imprinted films due to their different affinity abilities towards Co(II) ions. These findings are in agreement with those obtained from the LF isotherms for each developed Co(II)-IIP and NIP film sensor (see Appendix A). In fact, IIP2 and IIP8 were the most responsive high-affinity sensors, exhibiting heterogeneous binding sites (n > 1) with faster kinetics rebinding (K_D_ of 0.06 and 0.028 nM, respectively). On the other hand, IIP3, IIP5, and IIP7 resulted in lower-affinity IIPs, showing quite similar Δ*i_MAX_* responses to the acquired pattern of NIPs. In conclusion, the optimisation methodology of sensor responses of different developed IIPs with respect to sensitivities and imprinted factors is crucial and requires the mutual interpretation of sensor responses to gain effective control of MIP production.

### 3.3. Electrochemical Preparation of Sensors

Figure 2 shows typical CVs of polymeric film formation with 30 cycles recorded in a 0.5 mM acetate buffer solution (pH = 5.0) during the electropolymerisation of o-AP on an SPCE’s surface in the absence (a) and in the presence of Co(II) ions (b).

The electropolymerisation reaction was quite different in the first CV scan between the two prepared sensors. The voltammetric peak at +0.3 V (vs. Ag reference) belongs to o-AP monomer oxidation, leading to a subsequent chain propagation reaction and then to polymer formation. The presence of Co(II) ions in the polymerisation mixture during Co(II)-IIP film preparation promotes the appearance of an anodic peak at +0.93 V, suggesting the interaction between the monomer and the target during electrosynthesis, in comparison to the results obtained for the NIP films. On both electrodes, the electrochemical process is irreversible. In subsequent cycles, the peak current decreases, indicating the cycle-by-cycle growth of an insulating polymeric film formed on the electrode surface. 

### 3.4. Electrochemical Behaviour of Co(II)-IIP and NIP Films

CV and EIS measurements were used to electrochemically characterise the formed polymeric films on the SPCE (Appendix A). The bare SPCE shows no electrochemical activity in MES buffer in the applied potential scans. The formation of the Co(II)-IIP film and NIP films sensors on SPCEs resulted in relatively different electroactive polymers. To be specific, a redox couple peak at −0.18 V and −0.37 V appeared on both the prepared sensors, indicating the presence of an electroactive polymer after the electropolymerisation process. Moreover, the electroactivity acquired by the Co(II)-IIP film polymer was larger than that observed for the NIP films, which means that the Co(II) ions are embedded in the imprinted film and promote electron transferring on the electrode surfaces. When the sensors were soaked in H_2_SO_4_ for 10 min to remove template ions and unreacted monomers, the oxidation and reduction peaks shifted towards more negative potentials, indicating a structural modification in the polymeric network after the elution treatment. This behaviour has already been explained in the literature [24], where the same eluent was selected to guarantee polymeric modification and target elution during treatment. Further information can be found in the Appendix A.

The obtained EIS spectra are presented as Nyquist plots in Appendix A. Basically, the bare SPCE showed a very fast electron-transfer kinetics reaction, with an R_ct_ of 925 Ω. After the electrosynthesis of the Co(II)-IIP and NIP films (Appendix A), the resistance increased, confirming the presence of a non-electroactive polymer. However, the increase in R_ct_ was remarkably higher in Co(II)-IIP films (8.22 × 10^5^ Ω) than NIP films (2.85 × 10^5^ Ω), which could be explained by the presence of Co(II) within the polymer network. When the modified electrodes were treated with 0.3 M H_2_SO_4_, a decreased resistance was observed for both cases (1.50 × 10^5^ Ω for the NIP film and 1.59 × 10^4^ Ω for the Co(II)-IIP film, respectively). The decrease was remarkably higher for the Co(II)-IIP film as the emptied cavities in the IIP allowed for the diffusion of the redox probe through the polymer, facilitating electron transfer to the electrode surface. At the same time, the slight variation at the NIP film after H_2_SO_4_ treatment washing could be interpreted as the removal of unreacted monomers. After 10 min of rebinding in 31 nM of Co(II) ion solution, the R_ct_ increased on the Co(II)-IIP sensor (3.04 × 10^4^ Ω) due to the partial rebinding of the analyte within the prepared cavities. The semicircles of the NIP film also changed after incubation (2.10 × 10^5^ Ω), owing to the non-specific adsorption by the NIP film of the metal ion.

### 3.5. Analytical Assessment of Developed Sensors

Sensor signals for both NIP and Co(II)-IIP film sensors have been investigated with varied concentrations of Co(II) ions (from 1.95 to 125 nM) in a standard solution dissolved in MES buffer. As shown in Figure 3, the Co(II)-IIP film exhibited higher sensitivity compared to its counterpart NIP film. This behaviour can be simply explained by the presence of target-mimic cavities in the Co(II)-IIP network, which is sensitive to rebind with the imprinted target. Similarly, the NIP film showed slight differences in current after interacting with the tested Co(II) ions at different concentrations, which indicates that the NIP film could not properly recognise Co(II) ions and the interactions were made as a result of non-specific adsorption of the target by the polymer. The optimised Co(II)-IIP film sensor revealed a high imprinted factor and the lowest K_D_ towards its target.

Calibration curves of depicted concentrations are reported in Figure 3c, showing the comparison of the acquired sensor signals of the Co(II)-IIP film and the NIP film, confirming the superior sensitive properties of the Co(II)-IIP film towards its target. Estimation of adsorption parameters after Langmuir–Freundlich isotherm fitting shows a K_D_ of 0.03 ± 0.01 nM (R^2^ = 0.990) and 0.17 ± 0.01 nM for the Co(II)-IIP film and NIP film sensors, respectively, confirming the higher-affinity (5.8 times more compared to NIP) properties of cavities in the Co(II)-IIP films. The imprinting factor (α) calculated by dividing the Δ*i*_MAX_ obtained from the LF isotherms for both Co(II)-IIP and NIP films was 3.54. The limit of detection (LOD), calculated as 3 s/m where s is the standard deviation of blank samples and m is the slope of the regression plot at lower tested concentrations, was estimated to be 1.6 ± 0.2 nM. To further characterise the imprinted sensor with respect to selectivity features, a series of metal ion interferents were chosen and tested in the same concentration range as the Co(II) ions. For these purposes, the electrochemical responses (DPV measurements) were recorded after the exposure of the solutions of increased concentrations (from 1.92 to 62.5 nM) containing Hg(II), Cd(II), Cr(III), Ni(II), Mn(II), and Zn(II) ions as potential interferents, chosen for their environmental concern and similar ion radius (Co (II) = 74 pm). For each of the interfering species, a calibration plot was generated and compared with that of the Co(II)-IIP film. For simplification, slopes in the lower concentration range (from 1.9 to 15 nM) were compared as a sensitivity criterion; the selectivity factor (β) was calculated as S_Co(II)_/S_int_, where S_Co(II)_ is the sensitivity of the IIP film in the presence of Co(II) and S_int_ is the sensitivity of the IIP film in the presence of the interferents, in the range between 1.92 and 15 nM, (Table 2). These findings confirm that the Co(II)-IIP film sensor is selective toward Co(II) because of its recognition units. 

Of particular note, the sensor exhibited low sensitivities towards all of the tested interferents compared to its target, ranging from 3 times to 32 times the magnitude with respect to Mn(II) and Cr(III) ions. In addition, the results demonstrate that Zn(II) ions (74 pm) do not fit to sensor cavities—maybe due to a different spatial fit within the structure of the formed cavities—whereas Cd(II) ions may interfere more than the others, even if its magnitude remains limited. The proposed sensor was also not affected by trace concentrations of emergent HMs such as Hg(II) and Cr(III) ions, demonstrating an eightfold superior sensitivity towards Co(II) ions. 

The reproducibility of the Co(II)-IIP sensor was studied by determining the DPV current responses of three different electrodes fabricated according to the same procedure. A relative standard deviation (RSD) of 7.14% at a concentration 15.62 nM revealed good reproducibility. The repeatability (daily life) of the Co(II)-IIP film sensor was also tested over nine consecutive measurements (2 h), leaving the electrode in open air between each experiment trial. The results show that the designed sensor has good repeatability with a satisfactory RSD of 4.8 ± 0.66% for the concentration 31.25 nM of Co(II). To check the reusability of the sensor after its calibration, the Co(II)-IIP sensor was incubated in H_2_SO_4_ 0.3 M for 15 min and tested again with lower concentrations of Co(II). As a comparison, 97% of the initial current responses were maintained after rebinding with 3.9 nM of ions, while 88% of sensor responses remained available after rebinding with 15 nM of ions. This achievement remains an acceptable result, taking into account the nature of the disposable electrodes. 

In the literature, we found very few publications related to electrochemical sensing of Co(II) using MIP receptors. Table 3 summarises the analytical properties based on different sensing principles of reported sensors for the quantification of Co(II) ions. Compared to the reviewed optical sensors, our Co(II)-IIP film shows better analytical performance in terms of its dynamic range of working concentration and LOD, confirming that the proposed sensor is suitable for cobalt detection in real samples. Compared with other voltammetric sensors, our prepared electropolymerised IIP films can be proposed as a simpler route for preparing sensors based on MIPs, as opposed to the traditional ones [26], showing similar analytical performances with a reduced quantity of chemicals, cost, and time of polymerisation. In addition, the electrochemiluminescence sensing principle of an electropolymerised IIP combined with nanomaterials [27] provided similar results to those of our simple procedure of IIP synthesis, without requiring further modification during receptor preparation. 

Finally, the Co(II)-IIP sensor serves as a miniaturised, easy-to-produce, fast, and reliable device, useful for the daily analysis of complex matrix samples and for overcoming the challenge of regular water monitoring. 

### 3.6. Real Sample Analysis

As proof of application, the practicability of the Co(II)-IIP film sensor in detecting Co(II) ions was assessed by using spiked buffered aqueous samples. Initially, a volume of each blank sample solution (50 μL) was placed on the sensor’s surface to acquire DPV measurements. After monitoring the Co(II)-IIP film sensor’s response to the matrix, the prepared diluted samples were spiked with varied known concentrations and incubated on the electrode surface for 5 min. Finally, DPV measurements were performed, and recovery values were calculated (Table 4). 

The results in Table 4 show that in accordance with the acquired DPV signal, the matrix’s influence was minimal, and the recovery percent ranged from 99.3% to 106% in the spiked sea water and from 98% to 102.8% in the spiked tap water, confirming the practicability of the proposed sensor in quantifying Co(II) ions in real aqueous samples.

## 4. Conclusions

The present work describes the development of an IIP film-based electrochemical sensor for Co(II) recognition. The sensor was fabricated through electropolymerisation of 2-aminophenol in the presence of Co(II) at variable concentration ratios on the SPCE’s surface. The Taguchi optimisation model was employed to investigate the influence of IIP parameter conditions on sensor design, such as electropolymerisation cycles, the concentration of 2-aminophenol, the concentration of the target ion, and the extraction conditions in an acidic solution. Under the optimised conditions, the prepared sensor exhibited a K_D_-dependent rebinding capacity with good selective recognition of the target (IF = 3.54). The linear range was between 1.9 and 15 nM. Langmuir–Freudlich binding isotherm calibration plots were also investigated for pattern recognition under different polymeric electrosynthesis conditions, resulting in being essential for predicting the high-affinity preparation of IIPs compared to related NIP films. Selective features of the developed sensors with high reproducibility make the proposed device suitable for point-of-need monitoring of water matrices.

## Figures and Tables

**Figure 1 nanomaterials-14-00536-f001:**
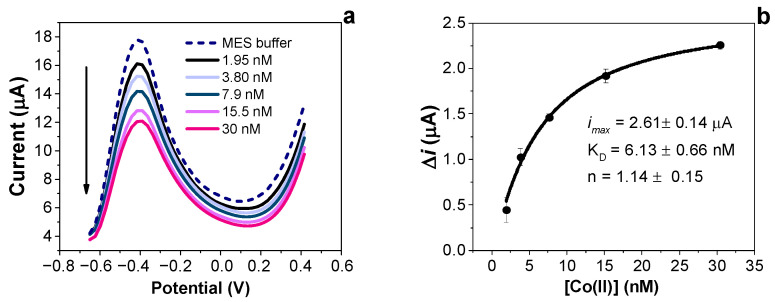
(**a**) Differential pulse voltammetric responses of the non-optimised Co(II)-IIP film to increased concentrations of Co(II) ions in MES buffer; (**b**) calibration plot and kinetics parameters obtained for the non-optimised Co(II)-IIP film.

**Figure 2 nanomaterials-14-00536-f002:**
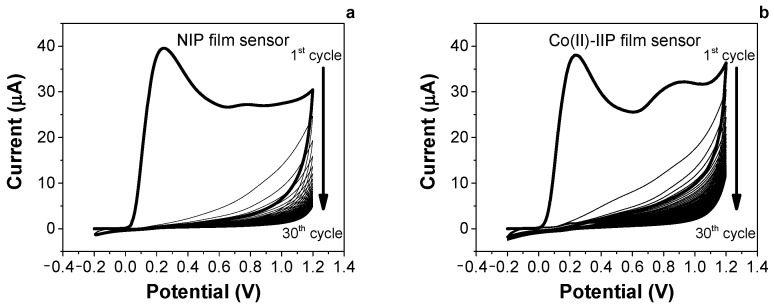
(**a**) CV of electropolymerisation of 0.5 mM o-AP during NIP film sensor preparation; (**b**) CV of electropolymerisation of 0.5 mM o-AP in the presence of 3 mM Co(II) ions during Co(II)-IIP film sensor preparation. Media: AB buffer (0.05 M, pH 5). Voltammetric conditions included potential range: from −0.2 to +1.2 V, scan rate: 50 mV s^−1^, and CV cycles: 30.

**Figure 3 nanomaterials-14-00536-f003:**
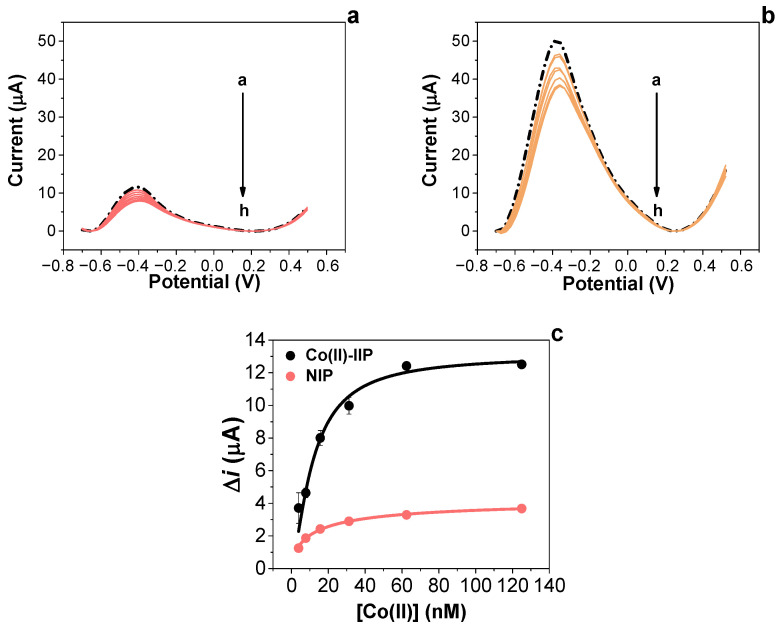
DPV measurements on (**a**) NIP film and (**b**) Co(II)-IIP film sensors after exposure to (a, dashed black line) 0, blank in MES buffer pH 5, 50 mM, (b) 1.95, (c) 3.90, (d) 7.81, (e) 15.62, (f) 31.25, (g) 62.5, and (h) 125 nM of Co(II) ions dissolved in MES buffer standard solution; (**c**) calibration plots comparing Co(II)-IIP and NIP films fitted with LF isotherm.

**Table 1 nanomaterials-14-00536-t001:** Level of selected parameters.

Parameters	Level 1	Level 2	Level 3
2-AP concentration, mM	0.2	0.5	1
Co(II) concentration, mM	1	2	3
CV scans	10	20	30
Elution time, min	10	25	40

**Table 2 nanomaterials-14-00536-t002:** Selectivity factors (β) calculated against potential ion interferents in the range of concentration between 1.92 and 15 nM.

Ion	Sensitivity (µA/nM)	Error, ± (nM)	Selectivity Factor, β
Co(II)	0.41	0.04	/
Mn(II)	0.11	0.02	3.8
Zn(II)	0.05	0.01	7.5
Hg(II)	0.06	0.03	6.8
Cr(III)	0.01	0.002	31.5
Cd(II)	0.11	0.01	3.6
Ni(II)	0.05	0.01	8.1

**Table 3 nanomaterials-14-00536-t003:** Comparison of analytical performances of proposed sensor with other electrochemical and optical sensors for Co(II) ion monitoring in water.

Sensor Configuration	Sensing Principle	Linear Range (nM)	LOD (nM)	Reference
MNPs-IIP ^a^	CSDPV	0.5–20 20–500	0.1	[26]
MWCNT/Cu/CQDs-IIP ^c^	ECL	1–100	0.31	[27]
S QDs	Fluorescence	0–9.0 × 10^4^	20	[28]
Fiber-QDs	Luminescence	0–3 × 10^6^	1 × 10^5^	[29]
Si QDs	Fluorescence	1 × 10^3^–1.2 × 10^5^	370	[30]
Co(II)-nitroso-S complex ^b^	ASV	55–3.2 × 10^3^	30	[31]
Co(II)-IIP film ^d^	DPV	1.9–15	1.6	This work

Transducer elements: ^a^ GCE: glassy carbon electrode; ^b^ CPE: carbon paste electrode; ^c^ AuE: gold electrode; ^d^ SPCE: screen-printed carbon electrode. Methods: DPV: differential pulse voltammetry; ASV: adsorptive stripping voltammetry; CSDPV: cathodic stripping differential pulse voltammetry; ECL: electrochemiluminescence. Sensing materials: Si QDs: silicon–carbon dots; MNPs: magnetic nanoparticles; IIP: ion-imprinted polymer; S QDs: sulphur quantum dots; MWCNT/CuNPs/CQDs: multiwalled carbon nanotube/Cu nanoparticles/carbon quantum dots.

**Table 4 nanomaterials-14-00536-t004:** Recovery test values of Co(II)-IIP film sensor applied to spiked real aqueous samples.

Sample	[Co(II)] Added, nM	[Co(II)] Found ±, nM	Recovery (%)
Tap water	3.9	4.0 ± 0.09	103
	7.8	7.7 ± 0.2	98
Sea water *	7.8	8.3 ± 0.4	106
	15.6	15.5 ± 0.2	99

* Sample validated according to ref. [20].

## Data Availability

Data will be shared upon request.

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
