# Peer review of "Rational Design of a Portable Chemometric-Assisted Voltammetric Sensor Based on Ion-Imprinted Polymeric Film for Co(II) Determination in Water"

_nanomaterials, 2024, doi:10.3390/nano14060536_

Round 1

Reviewer 1 Report

Comments and Suggestions for Authors

In this manuscript, IIP layers were synthesized on the surface of carbon electrode to realize the selective detection of Co(II). This work gives some useful results and information about the analytical means based on ion-imprinted polymer. In fact, as there is a lack of structural characterization, the content of this work can not be related to ‘Nanomaterial’. Other comments are given below:

1. A thorough proofreading is needed to eliminate all the minor grammatical errors, as some English expression details should be paid attention to, such as the correct use of ‘separately’ versus ‘respectively’, ‘compared with’ versus ‘compared to’, ‘owing to ‘ versus ‘due to’, ‘on the basis of’ versus ‘based on’, and ‘synthesis procedure/parameters’ versus ‘synthetic procedure/parameters’; and other grammatical issues also need to be concerned, e.g. ‘high stable’ should be corrected as ‘highly stable’, ‘KHz’ should be expressed as ‘kHz’;

2. In Figure 1a, the presence of Co(II) affects the reduction peak current at about -0.4 V vs. Ag. Is this trend reversible after the electrode is washed to remove Co(II)?

3. When discussing Table 2, the authors state that Zn(II) ions (74 pm) do not fit to sensor cavities. As Zn(II) is the same as Co(II) in ion radius, what is the reason for the difference in cavity affinity?

4. Can a linear response range be determined from Figure 3c, as this value is important to sensor? The results in Table 3 give a linear range of 1.92–125 nM, this is clearly wrong when Figure 3c is considered.

Comments on the Quality of English Language

Many expressional details should be corrected.

Author Response

We would like to thank the reviewer for comments, suggestions and observations. Please find attached our response. 

Reviewer 2 Report

Comments and Suggestions for Authors

Sabrina Di Masi and Cosimino Malitesta et al. present an exploration of chemometric assisted synthesis of electrochemical sensors based on electropolymerised ion imprinted polymeric (e-IIP) films. The study focuses on the development of Co(II)-IIP sensors using electropolymerisation procedures, following a Taguchi L9 experimental design. The synthesized e-IIPs are compared to their respective NIP films and evaluated based on binding kinetics, imprinting factor, and heterogeneity index. The results suggest that the proposed method can provide a generic protocol for synthesizing high-affinity, stable, and reproducible imprinted polymeric films for monitoring purposes. Experiments were well designed, and manuscript can be accepted after addressing the following concerns:

1.     The introduction did not provide enough information and could be a little bit confusing. What is IF. More description and background on MIPs should be added.

2.     Why “they” for your own works on line 61.

3.     The manuscript should be carefully proofed, for too many clerical errors, such as HMI and HM on Line 30, sentences on Line 101-102, and H2SO4 on Line 109.

4.     Full name or definitions should be added for the abbreviation for the first time, including the abstract. E.g. NIP on Line 17, CV, EIS on Line 92 and LF on Line 127. Please check the rest.

5.     Please check sentence on Line 165-166. What is blank?

6.     For the “Electrochemical preparation of sensors”, more characterization should be added to reveal the real morphology, including AFM, SEM and XPS. AFM and SEM for surface cavities, XPS for residual Co ions.

7.     Please explain the definition of LOD on Line 305. Why use this definition?

8.     How is Selectivity factor, β in Table 2 calculated?

Comments on the Quality of English Language

The language should be carefully proofed in academic manner.

Author Response

Thanks to the second reviewer. We carefully check all the manuscript following the suggestions of the reviewer. Please find attached our responses to each comment.

Round 2

Reviewer 2 Report

Comments and Suggestions for Authors

6. For the “Electrochemical preparation of sensors”, more characterization should be added to reveal the real morphology, including AFM, SEM and XPS. AFM and SEM for surface cavities, XPS for residual Co ions.

R6. Thank you for your suggestions. Actually we tried to perform SEM analysis on sensors but results were not significant improved the manuscript since it is a bit difficult to analyse screen printed electrodes properly. For other characterisation steps, we were not able to perform them.
Comments:  What do the authors mean by " results were not significant improved the manuscript"?  The obtained SEM results need to be shown in the supporting information at least and related analysis or discussion on why can not get the ideal results should be provided. Moreover, I still think XPS or related element characterization is necessary, for the preparation of the sensors, for the residual Co2+ could also effect the detection results.

Author Response

Dear Reviewer, 

please find enclosed our attachments.

Your sincerely, 

Sabrina Di Masi, Ph.D.

Round 3

Reviewer 2 Report

Comments and Suggestions for Authors

I am afraid I did not find "Fig. SM3B" in every version of the supporting materials. There are only 2 figures there. By the way, can the author include the figures in the reply, please? Moreover, I still recommend the authors to perform element characterization to check if there are residual ions, even EDS of SEM with proper comparison is acceptable.

Author Response

Dear Reviewer,

please find enclosed our reply to your comments.

The corresponding author, 

Sabrina Di Masi

Round 4

Reviewer 2 Report

Comments and Suggestions for Authors

The manuscript can be accepted as it is.

Author Response

Thank you for your comments and suggestions.